# DDIL: Improved Diffusion Distillation with Imitating Learning

## Abstract

Diffusion models excel at generative modeling (e.g., text-to-image) but sampling requires multiple denoising network passes, limiting practicality. Efforts such as progressive distillation or consistency distillation have shown promise by reducing the number of passes at the expense of quality of the generated samples. In this work we identify co-variate shift as one of reason for poor performance of multi-step distilled models from compounding error at inference time. To address co-variate shift, we formulate diffusion distillation within imitation learning (*DDIL*) framework and enhance training distribution for distilling diffusion models on both data distribution (forward diffusion) and student induced distributions (backward diffusion). Training on data distribution helps to diversify the generations by *preserving marginal data distribution* and training on student distribution addresses compounding error by *correcting covariate shift*. In addition, we adopt reflected diffusion formulation for distillation and demonstrate improved performance, stable training across different distillation methods. We show that DDIL consistency improves on baseline algorithms of progressive distillation *(PD)*, Latent consistency models *(LCM)* and Distribution Matching Distillation *(DMD2)*.

## 1 Introduction

Diffusion models, while capable of producing high-quality images, suffer from slow sampling times due to their iterative denoising process. To address this, distillation techniques have been proposed to reduce number of denoising steps. These techniques can be broadly categorized into trajectory-level (Luo et al., 2023; Meng et al., 2023; Salimans & Ho, 2021; Song et al., 2023) and distribution-matching approaches (Yin et al., 2023; 2024; Luo et al., 2024; Sauer et al., 2023) . While the former focuses on preserving the teacher's trajectory at a per-sample level, the latter matches the marginal distribution.

Multi-step student models offer a promising approach in balancing quality and computational efficiency. However, they often face a critical challenge: covariate shift. This occurs when the distribution of noisy input latents that the student model encounters during training differs from that seen during inference by the student. This mismatch can significantly impact generation quality especially when the number of denoising steps are low. Recent works Kohler et al. (2024); Yin et al. (2024) only consider backward trajectories to obtain feedback on quality of generation, but these approaches are often agnostic to data distribution and can exhibit mode-collapse.

In this work, we identify 'covariate shift' as a critical factor that impacts the generation quality in multi-step distilled diffusion models. To address covariate shift and to preserve diversity, we introduce diffusion distillation within the imitation learning (DDIL) framework by improving the training distribution for distillation. We achieve this by incorporating both the data distribution (forward diffusion) and the student's predictive distribution (backward trajectory at inference time). This approach combines the benefits of *(1) Preserving Marginal Data Distribution:* Training on the data distribution ensures the student model maintains the inherent statistical properties of the original data, and *(2) Correcting Covariate Shift:* Training on backward trajectories enables the student model to identify and adapt to covariate shifts, thereby improving the accuracy of score estimates, particularly in few-step settings. We illustrate instantiation of DDIL framework in context of progressive distillation in Figure 2. To this end, we make the following contributions:

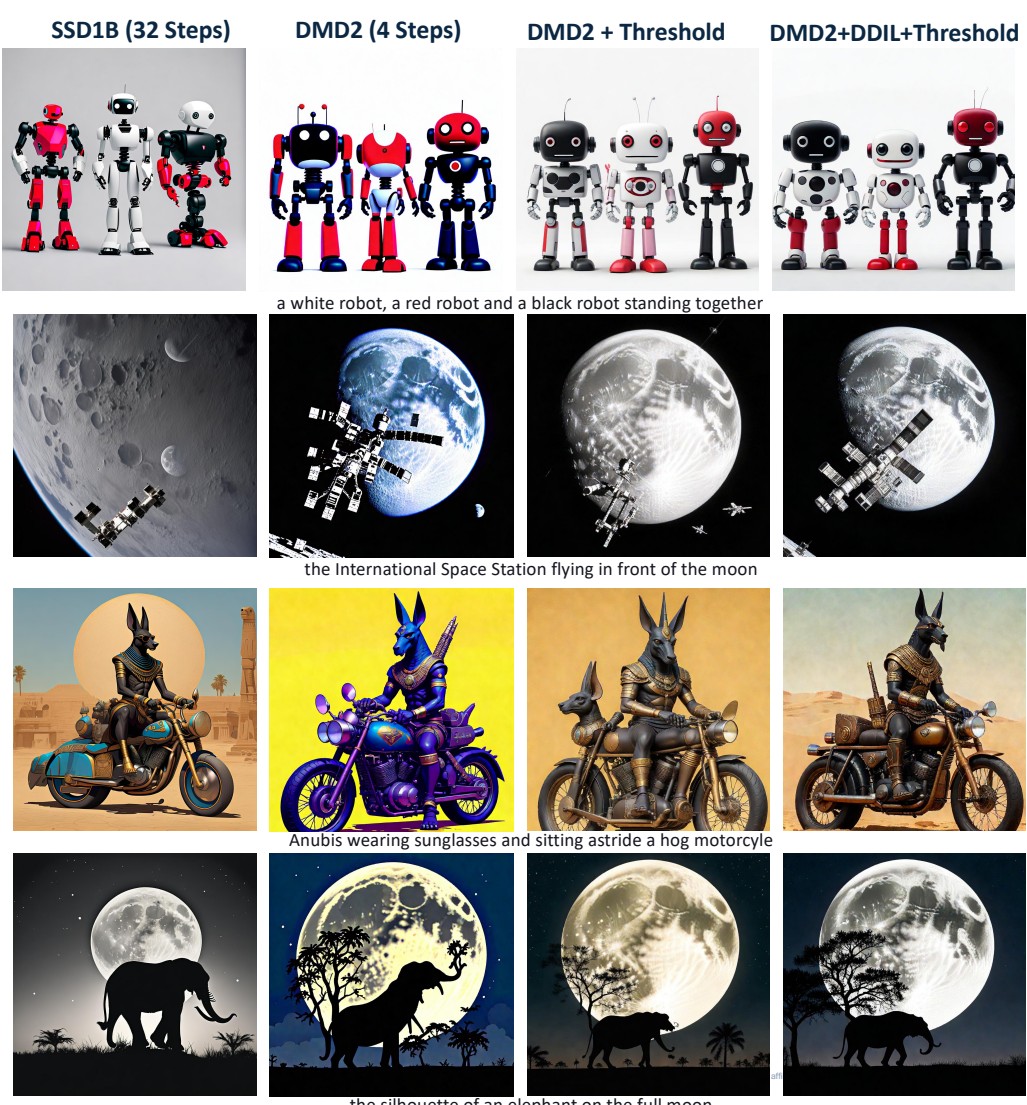

Figure 1: Qualitative comparison of images generated with different distillation techniques.

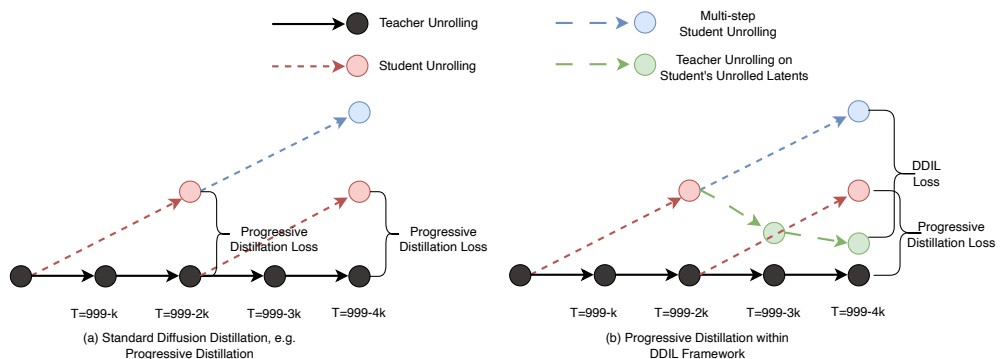

Figure 2: **Preditions at different timesteps for different distillation frameworks:** *(a)* We demonstrate standard progressive distillation training framework where student always sees forward diffused latent. *(b)* We show unrolling within our framework which in addition to (a) also obtains distillation feedback by querying teacher (green) on backward trajectory.

- We propose a novel *DDIL* framework which enhances training distribution of the diffusion distillation within the dataset aggregation 'DAgger' framework by performing distillation on on both the data distribution (forward) and student induced distribution (backward trajectory), yielding improved aggregate predictive distribution and better coverage.
- To enhance the stability of the distillation process in diffusion models, we adopt thresholding for both the teacher and student diffusion models to enforce the support of the data distribution with reflected diffusion Lou & Ermon (2023) for distillation. Consequently, this approach further mitigates covariate shift, leading to more substantial improvements when combined with DDIL
- We demonstrate that our DDIL approach yields diverse samples and consistently improves on different distillation techniques like progressive distillation (PD), latent consistency distillation (LCM), distribution matching distillation (DMD2) in a computationally efficient framework.

## 2 RELATED WORK

**Diffusion distillation methods.** Progressive distillation Salimans & Ho (2021); Meng et al. (2023) and many follow up works Li et al. (2023b); Berthelot et al. (2023) try to reduce the number of iterations of student model by forcing student to mimic multiple steps of the teacher. Consistency models Song et al. (2023); Luo et al. (2023); Ren et al. (2024) assume deterministic probabilistic flow at inference and enforce consistency in the data space for step-distillation. Additionally, recent work decomposes the diffusion trajectory into multiple segments like in progressive distillation and performs distillation within consistency formulation Kim et al. (2023). Instead of using the real data, methods such as BOOT Gu et al. (2023) consider bootstrapping in the student trajectory to generate samples of high quality and diversity. Liu et al. (2023) approximates the underlying map of the pretrained diffusion model as linear paths. While above trajectory level distillation techniques like progressive distillation and consistency-based approaches improve efficiency, the quality of the generated samples exhibits low visual fidelity.

Alternatively, diffusion distillation has been formulated in the distributional matching framework Yin et al. (2023); Luo et al. (2024); Yin et al. (2024); Salimans et al. (2024); Sauer et al. (2023; 2024). Within distribution matching approaches instead of matching teacher for each trajectory or particle like in previous class of methods, we try to match marginals of distilled student model and pretrained diffusion model. Further, adversarial loss has been applied to distillation approaches to improve the visual quality of the generated images Sauer et al. (2023; 2024); Lin et al. (2024). Most of distributional matching objectives like Sauer et al. (2024); Yin et al. (2023) are mode-seeking and looses on diversity. EM distillation Xie et al. (2024) addresses this by richer sampling with langevian MCMC to provide better target for distillation.

**Reverse diffusion as Markov decision process.** Policy gradient methods have recently gained traction in text-to-image generation with diffusion models by formulating the reverse diffusion process as a markov decision process (MDP) Fan et al. (2024); Xu et al. (2023a). Recent work Fan et al. (2024) proposes a policy gradient method for data distribution matching in diffusion models. Black et al. (2023) introduces a policy gradient algorithm with reward function that optimizes a diffusion model for downstream tasks. Yang et al. (2024) assumes a latent reward function of the reverse denoising process by emphasing the text and image alignment on the coarser steps of image generation. All these approaches have been applied to improve the alignment between the prompts and generated images for high-fidelity synthesis. In our work, we leverage the formulation of reverse process as MDP for step-distillation. This formulation allows interactively update the student model with the observations of the teacher model using dataset aggregation Ross et al. (2011).

## 3 BACKGROUND

### 3.1 REVERSE DENOISING PROCESS AS MDP

In imitation learning, an agent learns to perform tasks by observing and mimicking the behavior of the expert. An MDP in imitation learning models the next action based on the previous action and the current knowledge of the environment (Ke et al., 2021; Spencer et al., 2021). In general, an MDP is

represented as $\langle \mathcal{S}, \mathcal{A}, P, \rho_0 \rangle$, where $\mathcal{S}$ is a finite set of states, $\mathcal{A}$ is the set of actions, $P(s'|s, a)$ is a state transition kernel to transition from $s$ to $s'$ under the action $a$ and $\rho_0$ is the set of initial states. An MDP produces a trajectory which is a sequence of state-action pairs $\tau = (s_0, a_0, s_1, a_1, ..., a_T, s_T)$ over $T$ time steps.

We formalize the reverse process of the diffusion models as a finite horizon MDP (Black et al., 2023; Fan et al., 2024) with the policy $\pi_\theta$ (the diffusion model with parameters $\theta$) where the states and the actions are $s_t := (\mathbf{x}_t, t)$ and $a_t := \mathbf{x}_{t-1}$ respectively. The transition dynamics is defined by $P(s_{t+1}|s_t, a_t) := \delta(\mathbf{x}_{t-1}, t)$ and $\rho_0(s) := (\mathcal{N}(\mathbf{0}, \mathbf{I}), T)$ denotes the initial state distribution. The trajectory $\tau$ becomes $(\mathbf{x}_T, \mathbf{x}_{T-1}, \ldots, \mathbf{x}_0)$.

## 3.2 CO-VARIATE SHIFT IN DIFFUSION MODELS

With in iterative denoising steps of generation within backward trajectory of diffusion models, student's current predictions determines what the student (learner) sees in next step within sequential setting, which is classic feedback loop [1] in imitation learning. So if student makes any mistake or has bad score estimate in one of early steps this discrepancy excarbates in later iterations and results in accumulation of error. This error results in change in input distribution (covariate shift) of latents between training time (forward diffusion) and latents student model encounters when it is unrolled in iterative fashion at generation i.e., backward trajectory. Exposure bias is another closely related line of work Li et al. (2023a) which also discusses change in input distribution w.r.t pretrained diffusion model and propose training-free methods to improve it. Our work primarily focuses on distilling diffusion models and how this shift effects distillation.

Covariate shift is more prounced for distilled student diffusion model compared to pretrained diffusion model. To further clarify why covariate shift poses more of a challenge for the student model compared to the teacher model, we can consider inference as ancestral sampling (or annealing in score estimation). During generation i.e., within intermediate time-steps of backward trajectory of diffusion model, there is an implicit assumption that the marginal distributions between two consecutive denoising steps significantly overlap. While this is a reasonable assumption in continuous time diffusion models or when the number of denoising steps is sufficiently high for pretrained diffusion model, when considering a diffusion model with only few steps, this assumption does not hold. Consequently, any covariate shift would be more exacerbated for the student model, unlike the continuous time teacher mode.

**DAgger to mitigate Co-variate shift:** Imitation learning has long been used to learn offline sequential tasks wherein a student model is trained from teachers' demonstrations. Standard imitation learning also suffers from covariate shift in discrepancy in states visited by student and the teacher. Interactive methods such as DAgger (Ross et al., 2011) in Imitation learning augment training data by querying the teacher model on student's states, thereby obtaining teacher's feedback on student's predictive distribution(backward trajecories). Building on the ideas of interactive methods in imitation learning, in our work we aim to improve training distribution for diffusion distillation.

**Imitation Learning as Distribution Matching:** Notably, Ke et al. (2021) has shown that the imitation learning algorithms can be formalized as $f$-divergence minimization algorithms where the DAgger approach minimizes the total variation (TV) distance between the student and the teacher trajectory distributions. If $\rho_\theta(\tau)$ and $\rho_\eta(\tau)$ are teacher and student trajectory distributions respectively, then DAgger minimizes upperbound on total variation. DAgger achieves $O(T\epsilon)$ error compared to behavior cloning equivalent progressive distillation or any method with teacher forcing bounded by $O(T\epsilon^2)$ error.

$$D_{TV}(\rho_\theta(\tau), \rho_\eta(\tau)) \leq T\mathbb{E}_{s\sim\rho_\eta(s)}\left[D_{TV}(p_\theta(\mathbf{x}_{t-1}|\mathbf{x}_t, t), p_\eta(\mathbf{x}_{t-1}|\mathbf{x}_t, t)\right] \qquad (1)$$

## 3.3 BACKWARD TRAJECTORIES FOR DISTILLING DIFFUSION MODELS

Backward trajectory distillation introduced in recent concurrent works like ImagineFlash(Kohler et al., 2024) and DMD2 (Yin et al., 2024), focuses solely on evaluating the quality of generated samples without considering the data distribution. Consequently, they lack a mechanism to prevent mode collapse and ensure diversity.

Table 1: Properties of Different Diffusion Distillation Techniques

| Model | $x \sim p_{data}(x)$ | $x \sim q_\eta(x)$ | Preserve Diversity |
|---|---|---|---|
| Progressive Distillation (Meng et al., 2023; Salimans & Ho, 2021) | ✓ | ✗ | ✓ |
| ImagineFlash (Kohler et al., 2024) | ✗ | ✓ | ✗ |
| LCM | ✓ | ✗ | ✓ |
| InstantFlow (Liu et al., 2023) | ✓ | ✗ | ✓ |
| ADD (Sauer et al., 2023) | ✓ | ✓ | ✗ |
| DMD (Yin et al., 2023; 2024) | ✗ | ✓ | ✗ |
| DDIL (Ours) | ✓ | ✓ | ✓ |

**Mode Seeking:** The problem of covariate shift and distribution changes in diffusion distillation is multifaceted. It's not just about input distribution shifts caused by error accumulation. The reduction of diversity in the intermediate steps of backward trajectories (generative process) also plays a crucial role. If diversity is lost early on, it cascades through subsequent steps, limiting the range of possible outcomes. This is akin to error accumulation, but instead of errors, we are consistently losing diversity across time. We can think of it like sequential Monte Carlo sampling in diffusion models: at each step, we are discarding a large number of potential paths (particles), leading to a narrower range of possibilities in the later stages.

While EM Distillation (Xie et al., 2024) addresses this by employing Langevin MCMC for a richer reverse process and mode-covering divergences, it still doesn't explicitly incorporate the data distribution into its sampling prior during distillation.

Table 1 provides a summarized overview of the design choices adopted by different techniques.

## 4 METHOD

### 4.1 IMPROVING TRAINING DISTRIBUTION WITH DDIL

We introduce Diffusion Distillation with Imitation Learning (DDIL), a novel framework inspired by the DAgger algorithm from imitation learning to enhance the sampling distribution of intermediate noisy latents for distilling diffusion models. Diffusion model distillation involves two key considerations: (1) the training distribution of latent states encountered by the student model, and (2) the feedback mechanism employed during distillation. DDIL specifically focuses on improving the training distribution, remaining agnostic to the specific feedback mechanism utilized by different distillation techniques.

To achieve this, DDIL strategically samples intermediate latent variables from three sources: (1) forward diffusion of the dataset, captured by the sampling prior $\beta_{frwd}$ (as illustrated in Algorithm 1); (2) backward trajectories (unrolled latents) from the student model, denoted by the sampling prior $\beta_{student\_bckwrd}$; and (3) backward trajectories from the teacher model, denoted by the sampling prior $\beta_{teacher\_bckwrd}$, which is particularly advantageous in data-free settings for preserving marginal data distribution. Combining these sampling strategies leads to improved distillation performance.

DDIL is a unified training framework for distilling diffusion models w.r.t sampling prior for distillation. DDIL incorporates teacher feedback on student trajectories, aligning with the principles of DAgger Ross et al. (2011) in case of progressive distillation and latent consistency models (LCM). Furthermore, while methods like Kohler et al. (2024); Yin et al. (2024) perform distillation only on backward trajectories and don't account for marginal data distribution during distillation, DDIL addresses this by consistently incorporating feedback from the chosen distillation algorithm on both forward and backward trajectories. Our flexible framework thus allows for improved training distribution to boost the performance of diffusion distillation methods.

Algorithm 1 outlines a generalized framework for Diffusion Distillation with Imitation Learning (DDIL). This framework leverages a pre-trained diffusion model (teacher) and a student diffusion model, typically initialized with the teacher's parameters. Additionally, access to real data is assumed, providing representative samples from the marginal data distribution during the distillation process. The framework necessitates specifying hyperparameters for both the teacher and student models, including their respective discretization schemes. For simplicity we assume DDIM solver in 1. Distillation proceeds by randomly selecting one of three methods for sampling intermediate noisy

---

**Algorithm 1** Generalized DDIL framework for Distilling Diffusion Models

---

**Require:** Teacher diffusion model with text-conditioning with params: $\theta$; student parameters: $\eta$; Dataset $\mathcal{D}$; Time step Discretization $N, N_s$ of Teacher and student Models respectively.

$k = 1000/N$          $\triangleright$ step size teacher diffusion model

$k_s = 1000/N_s$          $\triangleright$ step size of student diffusion model

$x \sim D$          $\triangleright$ Sample from data

$T_s \sim \{1000, 999, \ldots, 1\}$          $\triangleright$ Sample time-step

$\epsilon \sim \mathcal{N}(\mathbf{0}, \mathbf{I})$          $\triangleright$ Sample noise

         $\triangleright$ Choose current mini-batch sampling mode $\sim$ [forward, teacher backward, student backward]

**if** $p \sim U[0,1] < \beta_{frwd}$ **then**          $\triangleright$ Forward Process

     $z_t = \alpha_t x + \sigma_t \epsilon$          $\triangleright$ add noise to data

     $z_{T_s} \leftarrow z_t$

**else if** $\beta_{frwd} \leq p < \beta_{teach\_bckwrd}$ **then**          $\triangleright$ Teacher Backward

     **for** $t = \{1000, 1000 - k, \ldots, T_s\}$ **do**

         $\mathbf{z}_{t-k} = \alpha_{t-k}(\alpha_t \mathbf{z}_t - \sigma_t \hat{\mathbf{v}}_t) + \sigma_{t-k}(\sigma_t \mathbf{z}_t - \alpha_t \hat{\mathbf{v}}_t)$

         $t \leftarrow t - k$

     **end for**

     $z_{T_s} \leftarrow z_{t-k}$

**else**          $\triangleright$ Student Backward $\beta_{student\_bckwrd}$

     **for** $t = \{1000, 1000 - k_s, \ldots, T_s\}$ **do**

         $\mathbf{z}_{t-k_s} = \alpha_{t-k_s}(\alpha_t \mathbf{z}_t - \sigma_t \hat{\mathbf{v}}^s_t) + \sigma_{t-k_s}(\sigma_t \mathbf{z}_t - \alpha_t \hat{\mathbf{v}}^s_t)$

         $t \leftarrow t - k_s$

     **end for**

     $z_{T_s} \leftarrow z_{t-k_s}$

**end if**

Train student diffusion model on $z_{T_s}$ with distillation method.

---

latent inputs to the student model. This selection is governed by user-defined sampling priors: $\beta_{frwd}$, $\beta_{teach\_bckwrd}$, and $\beta_{student\_bckwrd}$, which correspond to the three sources of intermediate latents previously discussed. The choice and updating of these sampling priors, denoted as $\beta_i$, can be tailored based on the training stage, objective function, and overall task goals.

Let $q_\eta$ be the predictive distribution of distilled student diffusion model from its generated backward trajectories, then overall DDIL objective is

$$L_{\text{DDIL}} = \mathbb{E}_{t,\epsilon,\tilde{\mathbf{x}} \sim p_{data}(\mathbf{x})} L_{\text{Distill}} + \mathbb{E}_{t,\epsilon,\tilde{\mathbf{x}} \sim q_\eta(\mathbf{x})} L_{\text{Distill}} \tag{2}$$

Where $L_{\text{Distill}}$ can assume any objective based on chosen algorithm like progressive distillation, latent consistency distillation and distribution matching objective and $L_{\text{DDIL}}$ trains on both data distribution and backward trajectories.

**Reflected Diffusion Distillation:** When distilling diffusion models either the teacher or student model might not satisfy implicit assumed support during distillation which could makes training unstable and require large batch sizes, etc. We adopt reflected diffusion models Lou & Ermon (2023) framework for distillation i.e., threshold teacher's score estimate and/or student's and improve stability of training, lower required batch size with improved performance.

Thresholding is applied to the teacher model's estimates consistently across all investigated methods: progressive distillation, Latent Consistency Models (LCM), and DMD2. Furthermore, within the consistency distillation framework, the target derived from the student model is also threshold-ed. For DMD2, thresholding is applied to the score estimates of the pre-trained diffusion model, the fake critic, and the student model. Without thresholding our gradient feedback could be noisy and negatively impacting training stability.

### 4.2 DDIL INTEGRATION

This section examines the integration of DDIL with various distillation techniques. Detailed design choices are further elaborated in the appendix (section to be updated).

**PD + DDIL:** DDIL is integrated with progressive distillation using a DAgger-inspired approach Ross et al. (2011). Distillation is performed on mixed rollouts generated by alternating between the

pre-trained and student diffusion models within each generation. A stateless DDIM solver facilitates this interleaved sampling process.

**LCM + DDIL:** DDIL is also applied to consistency distillation. Due to the pre-trained model's lack of prior consistency training, the mixed-rollout strategy used in progressive distillation is not directly applicable. Therefore, DDIL is extended to LCM by applying consistency distillation to both forward and backward trajectories of the student model, leveraging the student-induced distribution and demonstrating performance improvements.

**DMD2 + DDIL:** Mirroring the progressive distillation approach, mixed rollouts are employed within the DMD2 framework. Trajectories are sampled up to a predetermined noise level or timestep (e.g., t=500) using either the student or teacher model. The resulting latent serves as input to the student model for gradient feedback within the DMD2 formulation. Distribution Matching aligns well within our DDIL framework as discussed in Section 3.2 where within imitation learning framework we consider matching generated trajectory distributions of student model with teacher or expert's trajectory distribution (of states or equivalently noisy latents in case of diffusion).

## 5 EXPERIMENTS

**Datasets and metrics:** Following standard practice for evaluating text-to-image diffusion models Rombach et al. (2022); Meng et al. (2023), we evaluate our distilled models zero-shot on two public benchmarks: COCO 2017 (5K captions), and COCO 2014 Lin et al. (2014) (30K captions) validation sets. We use each caption to generate an image with a randomized seed and report CLIP score using OpenCLIP ViT-g/14 model Ilharco et al. (2021) to evaluate image-text alignment. We also report Fréchet Inception Distance (FID) Heusel et al. (2017) to estimate perceptual quality. To measure diversity of generation, we report $LPIPS_{Diversity}$, where for a given prompt we generate output for 10 different seeds and obtain pair-wise LPIPS score and finally average over 50 randomly sampled COCO 2017 prompts.

**Training:** For all our experiments, we choose AdamW optimizer Loshchilov & Hutter (2017) with $1e-05$ learning rate with warmup and linear schedule on a batch size of 224 in case of progressive distillation, 360 in case of LCM and 7 in case of DMD2 on SSD1B. To optimize for GPU usage, we adopt gradient checkpoint and mixed-precision training. Please refer to Appendix A for additional training details.

**PD + DDIL:** In case of progressive distillation, we train the model for 4k steps for $\epsilon$ to $v$ space conversion to perform step distillation in $v$ space Salimans & Ho (2021). Then we perform guidance conditioning following the same protocol as Meng et al. (2023) where we sample guidance scale $\omega \sim [2, 14]$ and incorporate additional guidance embedding as in Rombach et al. (2022) followed by step distillation. Overall we train $10K$ steps to obtain guidance conditioned checkpoint SD($gc$). For progressive distillation, we start with a 32-step discretization assumption for the pre-trained diffusion model and perform $32 \rightarrow 16$ step distillation for 5K iterations with 500 steps of warm-up. We progressively increase training compute or gradient steps as we go towards fewer iteration student. For $16 \rightarrow 8$, we follow a similar protocol, but we further split each stage of training into two parts. First, we do distillation for 6K steps to obtain a checkpoint and resume with warmup and $1e-05$ learning rate for another 4K steps of training. For $8 \rightarrow 4$ and $4 \rightarrow 2$ distillation, we first distill model for 8K steps followed by another 6K steps. We split single stage of training into two parts to exactly match training protocol of Step Distillation and DDIL. See supplemental for additional details. We adopt timesteps for discretization from the default config of the DPM++ solver i.e., for 4-step models our timesteps are $\{999, 749, 500, 250\}$.

**LCM + DDIL:** We trained both LCM and DDIL models on the Common Caption dataset for 8,000 steps, using the SDv1.5 checkpoint and a batch size of 60 on 6 A100 GPUs. To enhance consistency distillation, we introduced backward trajectory sampling. Specifically, we randomly selected a number of inference steps (3, 4, or 5) and obtained samples at specific timesteps along the backward trajectory. This enabled us to incorporate consistency distillation loss feedback not only on forward diffused latents but also on these backward trajectory latents within our framework.

**DMD2 + DDIL:** In this work we consider distilling SSD1B checkpoint Gupta et al. (2024) with DMD2 for computational efficiency. To achieve stable training within the DMD2 framework, which utilizes a teacher model and a "fake" critic, we update the fake critic ten times for every update of the

Table 2: Text guided image generation results on $512 \times 512$ **MS-COCO 2017-5K** validation set. Our 4-step Model demonstrates SOTA performance for checkpoints based on SD1.5 where as ADD is based SD2.1 with more expressive text encoder and LCM also uses different checkpoint. '∗' denote derived baselines and '‡' denote different checkpoints

| Model | Steps | NFEs | FID [↓] | CLIP [↑] | LPIPS$_{Diversity}$[↑] |
|---|---|---|---|---|---|
| SnapFusion* Li et al. (2023b) | 8 | 16 | 24.20 | 0.300 | N/A |
| Step Distillation* *et al.*Meng et al. (2023) | 8 | 8 | 26.90 | 0.300 | N/A |
| Step Distillation* *et al.*Meng et al. (2023) | 4 | 4 | 26.40 | 0.300 | N/A |
| UFOGen* *et al.*Xu et al. (2023b) | 1 | 1 | 22.5 | **0.311** | N/A |
| ImagineFlash* ‡$Kohler et al.$ (2024) | 2 | 2 | 34.7 | 0.301 | N/A |
| ADD ‡ | 1 | 1 | 19.7 | 0.328 | 0.52 |
| LCM ‡ | 4 | 4 | 36.36 | 0.294 | 0.49 |
| LCM-LoRA ‡ | 4 | 4 | 37.01 | 0.300 | 0.52 |
| LCM-LoRA | 4 | 4 | 36.46 | 0.291 | 0.61 |
| Instaflow(0.9B) Liu et al. (2023) | 1 | 1 | 23.4 | 0.303 | 0.61 |
| LCM | 4 | 4 | 24.39 | 0.305 | 0.61 |
| + Reflected | 4 | 4 | 24.25 | 0.306 | 0.59 |
| + DDIL | 4 | 4 | 23.44 | 0.308 | 0.59 |
| + Reflected +DDIL | 4 | 4 | 22.86 | **0.309** | 0.59 |
| Progressive Distillation | 4 | 4 | 23.34 | 0.302 | 0.60 |
| +DDIL | 4 | 4 | **22.42** | 0.302 | 0.60 |
| Progressive Distillation | 2 | 2 | 26.43 | 0.288 | 0.58 |
| +DDIL | 2 | 2 | 24.13 | 0.291 | 0.58 |
| SD ($v$) | 32 | 64 | 22.50 | 0.321 | 0.62 |
| SD ($gc$) | 32 | 32 | 24.46 | 0.304 | 0.62 |

student model. We present two sets of results. The first serves as an oracle experiment, establishing an upper bound on performance. Here, we distill SSD1B into the DMD2 student model using a batch size of 64, 70,000 gradient updates, and SDXL critics. We then compare this oracle experiment with SSD1B checkpoints trained with a significantly smaller batch size of 7 and 30,000 gradient steps on a single A100 node. Reducing the batch size drastically leads to unstable training and poor performance. However, by employing reflected diffusion distillation, we achieve improved training stability and a significant boost in performance, both quantitatively and qualitatively. Further performance gains are observed when incorporating a mixed rollout setting of DDIL within DMD2, as demonstrated in the table 3

**Student Selection Prior:** Our protocol for student selection in trajectory collection follows standard practice from imitation learning. Where early in training, student's performance is bad and hence we prioritize sampling more from $p_{data}(x)$ but as training progress and student's performance is good we want to obtain expert feedback on mistakes that student makes i.e., address co-variate shift caused by feedback and training, inference mismatch but still sample from $p_{data}(x)$ to preserve marginal data distribution.

## 5.1 Text-guided Image Generation

We demonstrate effectiveness of our proposed DDIL framework across different baseline distillation techniques in case of text-to-image generation tasks as shown in Table 2. DDIL consistency improves on progressive distillation(PD) and latent consistency models (LCM) as observed in Table 2/ In case of progressive distillation, for 4-step version DDIL improves FID from $23.34 \rightarrow 22.42$ and maintains clip score of $0.302$ and similarly we can also observe DDIL improves on LCM with FID from $24.25 \rightarrow 22.86$ and CLIP score $0.306 \rightarrow 0.309$. From Tab. 4 in appendix, we can observe that 4-step variant of $PD + DDIL$ with a guidance value of 8 achieves best FID of 13.97, the highest among trajectory based distillation methods.

We also demonstrate effectiveness of DDIL with distribution matching techniques which adopt multi-step student like in DMD2. When we apply DMD2 to SSD1B, we can observe that DDIL improves FID from $31.77 \rightarrow 27.72$ and clip score from $0.320 \rightarrow 0.326$ and HPSv2 score from $0.302 \rightarrow 0.304$.

**Computational efficiency:** DDIL demonstrates superior computational efficiency compared to state-of-the-art methods such as Instaflow and DMD. For instance, LCM augmented with DDIL (LCM+DDIL) achieves strong performance using only 8,000 gradient steps with a batch size of 420.

This contrasts sharply with Instaflow, which requires 183 A100 GPU-days for distillation. DDIL with progressive distillation (PD) reduces this to 15 A100 GPU-days. Similarly, while DMD necessitates 64 GPUs with a larger batch size and extended training duration, DDIL attains comparable results using significantly fewer resources.

To minimize the overall computational burden of distillation, all experiments utilizing DMD2 employed a single A100 node with a batch size of 7. For a direct comparison with the original DMD2 configuration, an oracle experiment is conducted, distilling SSD1B with DMD2 across four A100 GPU nodes with a batch size of 64 and 70,000 gradient updates. We observed that training DMD2 with a smaller batch size exhibited instability, even with 10 fake critic updates per student update. However, integrating reflected diffusion and DDIL yielded improved stability and performance, producing qualitative results comparable to the larger-scale experiment.

By incorporating the DDIL framework and the reflected diffusion distillation formulation, we demonstrate enhanced training stability and achieve strong performance with DMD2 and LCM using significantly smaller batch sizes and fewer gradient updates.

**Diversity vs. quality trade-off:** Adversarial distillation methods Sauer et al. (2023; 2024) exhibit a decrease in generation diversity (measured by $LPIPS_{Diversity}$) compared to the baseline. Ss discussed in previous section most common objectives of distribution matching also have mode-seeking objects Xie et al. (2024); Yin et al. (2024). This highlights the quality-diversity trade-off often encountered in generative models when fine-tuning for human preferences or specific applications.

Table 3: Text guided image generation results on $512 \times 512$ **COCO 2017-5K** validation set. These results are obtained by adopting latent consistency distillation retrained for **SSD1B** and incorpating DDIL within DMD2 setting. We integrate DDIL into the DMD2 framework by unrolling just the student model exactly like DMD2 but also unrolling teacher to corresponding noise level too to better capture underlying data distribution and align gradient fields of student model and teacher model.

| Model | Steps | FID [↓] | CLIP [↑] | HPSV2 [↑] | LPIPS$_{Diversity}$[↑] |
|---|---|---|---|---|---|
| SSD1B (guidance = 8) | 20 | 30.23 | 0.336 | 0.297 | 0.48 |
| SSD1B-LCM | 4 | 35.23 | 0.311 | 0.282 | 0.45 |
| SSD1B-DMD2 (batch = 64,70k) | 4 | **26.56** | **0.337** | **0.309** | 0.49 |
| SSD1B-DMD2 (batch = 7,30k) | 4 | 31.77 | 0.320 | 0.302 | 0.52 |
|     + Reflected (batch = 7,30k) | 4 | 29.32 | 0.323 | 0.302 | 0.51 |
|     + Reflected + DDIL (batch = 7,30k) | 4 | **27.72** | **0.326** | **0.304** | 0.51 |

# 6 CONCLUSION

This work introduces DDIL, a novel framework for distilling diffusion models that addresses the challenge of covariate shift while preserving the marginal data distribution. Integrating DDIL with established distillation techniques, including Progressive Distillation, Consistency Distillation (LCM), and Distribution Matching based Distillation (DMD2), consistently yields quantitative and qualitative improvements. Furthermore, we also show that integrating DDIL within the DMD2 framework enhances training stability, reduces required batch sizes, and improves computational efficiency demonstrating wider applicability and practical usefulness.

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
