

| SDv1.5 (50 Steps) | InstaFlow (1 Steps) | Step Distillation (4 Steps) | DDIL (4 Steps) |

high quality colored pencil sketch portrait of furry blue fox, handsome eyes, photo of notebook sketch

Astronaut on Mars during sunset

Panoramic view of mountains of Vestrahorn and perfect reflection in shallow water, soon after sunrise, Stokksnes, natural lighting

a hyper realistic photo of a beautiful cabin inside of a forest and full of trees and plants, with large aurora borealis in the sky

Figure 1: Qualitative comparison of images generated with different distillation techniques.

# DDIL: Distilling Diffusion Models with Imitation Learning

## – Supplemental Material –

## A    ADDITIONAL TRAINING DETAILS

In case of progressive distillation, we use DPM++ multi-step Lu et al. (2022) solver (2-step) for all evaluations except co-variate shift analysis, where a state-less solver like DDIM Zhang et al. (2022) will ease switching between two different models and corresponding reverse processes. We train our model on an internal text-to-image dataset and only perform distillation without updating the teacher model. In case of consistency distillation with LCM, we adopt the default LCMScheduler. In the case of DMD2, we follow the same settings as the original repository except that in our work, we focus on SSD1B Gupta et al. (2024) instead of SDXL to reduce computational requirements for training. By

Table 1: Text guided image generation results on $256 \times 256$ **COCO 2014** val set.

| Model | Steps | FID [↓] |
|---|---|---|
| LCM-LoRA (4-step) | 4 | 23.62 |
| LCM-LoRA (2-step) | 2 | 24.28 |
| DMD | 1 | 14.93 |
| Progressive Distillation (PD) | 4 | 14.72 |
|  +*DDIL* | 4 | **13.97** |
| Progressive Distillation (PD) | 2 | 16.46 |
|  +*DDIL* | 2 | 15.81 |
| SD | 50 | 13.45 |

default, we adopt reflected diffusion for all three diffusion distillation techniques we considered in this work.

We follow similar protocol as progressive distillation for DDIL but as we now distill using on text-to-image data using forward process and interactively mixed unrolled trajectories, we have this additional sampling prior hyper parameters. For $16 \rightarrow 8$ distillation $75\%$ we obtain $z_t$ using forward diffusion making this part of training exactly equivalent to step Distillation. For remaining $25\%$ we choose data from mixed interaction unrolled trajectories where for first 6K steps, student is only selected $15\%$ of time for $z_t \rightarrow z_{t-2k}$ transition for trajectory collection, where as for next 4K iterations we choose student $80\%$ of time and train $50\%$ of time on collected trajectories. Effectively for first part of training we have prior $x \sim p_{data}(x)$ as $0.75 * 0.85$ followed by $0.5 * 0.2$ for later part of training within a stage. And follow same sampling prior protocol for two parts of training for each stage of progressive distillation with same hyper-parameters as step distillation. Overall we need $40K$ updates to obtain a 4-step checkpoint or $55K$ updates to obtain 2-step checkpoint for step distillation or DDIL, as our sampling is parallelized across GPUs we observe $< 5\%$ overhead for DDIL over step distillation, it takes 2 days on single node of 8 A100 GPUs to perform distillation.

## B  ADDITIONAL EVALUATION RESULTS

In Table 1 we demonstrate that DDIL achieves best performance among trajectory based distillation methods.

## C  DIFFUSION DISTILLATION METHODS

### C.1  PROGRESSIVE DISTILLATION

Progressive Step distillation aims at reducing the number of timesteps $T$ of the sampling (reverse) process in the diffusion models by learning a new student model. Starting from timestep $t$ within the reverse diffusion process, given the discretization interval $k$, $N$ steps of the teacher are distilled into $N/k$ steps of the student ($T = N$ in the first iteration of step-distillation) Salimans & Ho (2021). We query the teacher model at timesteps $t - k$ and $t - 2k$ while the student estimates are obtained at $t - 2k$ using a DDIM Song et al. (2020).

Following the formulation in Li et al. (2023), the teacher model is unrolled for two DDIM steps to $t - k$ and $t - 2k$ starting at timestep $t \in [T]$ and $0 \le t - 2k < t - k$ with input noisy latent $\mathbf{z}_t$ while student model performs one denoising step. Where $\hat{\mathbf{v}}_t^s$ is the velocity estimate from the student model $\mathbf{v}_\eta(\hat{\mathbf{z}}_t, t)$. The student model predicts the latent $\mathbf{z}_{t-2k}^s$ from $\mathbf{z}_t$ of the teacher and thus $\mathbf{z}_{t-2k}^s = \mathbf{z}_{t-2k}$. Progressive distillation loss with guidance conditioned teacher is denoted by,

$$L_{\text{PD}} = \max\left(1, \frac{\alpha_t^2}{\sigma_t^2}\right) \left\| \hat{\mathbf{x}}_t^s - \frac{\mathbf{z}_{t-2k} - \frac{\sigma_{t-2k}}{\sigma_t}\mathbf{z}_t}{\alpha_{t-2k} - \frac{\sigma_{t-2k}}{\sigma_t}\alpha_t} \right\|_2^2. \tag{1}$$

Here, the student model is trained with teacher forcing as is evident in Equation (1). During sampling from the student model, the teacher observations are not provided and therefore, the student model can drift from the expected trajectory Huszár (2015).

---

**Algorithm 1** Interactive Trajectory collection for Dataset Aggregation (DAgger)

---

**Require:** Teacher diffusion model with text-conditioning with params: $\theta$; teacher velocity: $\mathbf{v}$; student model velocity: $\mathbf{v}^s$; student parameters: $\eta$
**Require:** Initialize DAgger dataset to collect trajectories, $\mathcal{D}_{DAgger} \leftarrow \emptyset$
**Require:** Student diffusion models with text-conditioning and parameters $\eta$
    $\mathbf{x}_T \sim \mathcal{N}(\mathbf{0}, \mathbf{I})$
    $t = T = 1000$
    $k = 1000/N$            ▷ step length under assumed discrete setting of current teacher diffusion model
    **for** $t = \{1000, 1000 - 2k, ..., 1\}$ **do**
        **if** $p \sim U[0, 1] < \beta$ **then**          ▷ Choosing Student model vs teacher model for current iteration
            # One step of student DDIM step
            $\mathbf{z}_{t-2k} = \alpha_{t-2k}(\alpha_t \mathbf{z}_t - \sigma_t \hat{\mathbf{v}}^s_t) + \sigma_{t-2k}(\sigma_t \mathbf{z}_t - \alpha_t \hat{\mathbf{v}}^s_t)^s$
        **else**
            # 2 steps of DDIM with teacher
            $\mathbf{z}_{t-k} = \alpha_{t-k}(\alpha_t \mathbf{z}_t - \sigma_t \hat{\mathbf{v}}_t) + \sigma_{t-k}(\sigma_t \mathbf{z}_t - \alpha_t \hat{\mathbf{v}}_t)$
            $\mathbf{z}_{t-2k} = \alpha_{t-2k}(\alpha_{t-k} \mathbf{z}_{t-k} - \sigma_{t-k} \hat{\mathbf{v}}_{t-k}) + \sigma_{t-2k}(\sigma_{t-k} \mathbf{z}_{t-k} - \alpha_{t-k} \hat{\mathbf{v}}_{t-k})$
        **end if**
        $t \leftarrow t - 2k$
        Get trajectory $\tau_i = (\mathbf{z}_t, \epsilon, t)$ based on induced distribution of $\mathbf{v}, \mathbf{v}^s$
    **end for**
    Add $\tau_i$ to dataset, $\mathcal{D}_{DAgger} \leftarrow \mathcal{D}_{DAgger} \cup \tau_i$

---

## C.2 CONSISTENCY MODELS

While multi-step extensions of consistency distillation decompose the trajectory and enforce consistency within segments they remain susceptible to covariate shift with respect to the backward trajectory. This stems from the inherent discrepancy between the teacher and student model's perception of the data distribution in backward trajectory, which is addressed by our proposed DDIL framework. Hence, benefits of DDIL are complementary and extend to multi-step Consistency Distillation variants like CTM and TCD.

## D PROGRESSIVE DISTILLATION WITH DDIL

Inspired by the success of the interactive learning DAgger algorithm in imitation learning and following the formulation of the reverse process of the diffusion models as probability flow ODE, we first extend the DAgger framework to diffusion models by considering the higher iteration denoising model as expert and fewer iteration denoising model as a student in Algorithm 1 Following this, in Algorithm 2, we present the complete DDIL approach with interactive learning.

For sampling in diffusion models, the student predicted latent $\mathbf{z}_t$ is aligned with the teacher trajectory by adding the state-action pair $(\mathbf{z_t}, \epsilon, t) \in \tau_\theta$ to the aggregated dataset. Note that the dataset aggregation is done randomly so that the model is aware of the teacher and the student's distributions.

In our DDIL algorithm outlined in Algorithm 2, the distillation is performed iteratively by taking the sample from the aggregated dataset or from the default training dataset with forward diffusion. Following this, two steps of DDIM sampling are performed on the teacher model to obtain the estimate $\mathbf{z}_{t-2k}$ and subsequently optimize Equation (1). This framework introduces a self-correcting behavior. Even if the student deviates from the teacher's trajectory at any step of the reverse diffusion process – the teacher can provide corrective feedback.

## E CO-VARIATE SHIFT ANALYSIS

To validate our hypothesis of covariate shift from accumulation of error, we conduct a mixed-rollout evaluation using a 32-step CFG teacher and a 4-step DDIL-distilled student on the MS-COCO 2017 (5K) dataset. Both models achieve similar FID scores ( 22.5) with the teacher model having a CLIP score of 0.321. We assume a 32-step teacher model, where 1 student step is equivalent to 8 teacher steps. This allows for alignment between teacher and student estimates at specific timesteps in the diffusion process $999, 749, 500, 250$. This setting enables stochastic mixing between the teacher and student models during inference by choosing a state-less DDIM solver. We investigate three settings

**Algorithm 2** DDIL: Progressive Distillation on the aggregated dataset and forward diffusion within DDIL framework, assumes PF-ODE and deterministic sampling.

---

**Require:** Teacher diffusion models with text-conditioning and parameters $\theta$
**Require:** Data set $\mathcal{D}$
**Require:** Initialize DAgger dataset to collect trajectories, $\mathcal{D}_{DAgger} \leftarrow \emptyset$
**Require:** Number of teacher model denoising steps $N$
  **for** $L$ iterations **do**
      $\eta \leftarrow \theta$                                                        ▷ Initialize student from teacher
      $k = 1000/N$
      **while** not converged **do**
          $\#\mathbf{z}_t$ from from aggregated dataset or forward process
          **if** $p \sim U[0,1] < p$ **then**
             $(\mathbf{z}_t, \epsilon, t) \sim D_{DAgger}$                         ▷ sampled from mixed unrolling 1
          **else**
             $\mathbf{z}_t = \alpha_t \mathbf{x} + \sigma_t \epsilon$, where $x \sim D$, $t \sim U[0,1] * k$, and $\epsilon \sim \mathcal{N}(\mathbf{0}, \mathbf{I})$      ▷ Forward process
          **end if**
          $\hat{\mathbf{x}}_t^s = \alpha_t \mathbf{z}_t - \sigma_t \mathbf{v}_t^s$
          $\#$ 2 steps of DDIM with teacher
          $\mathbf{z}_{t-k}^* = \alpha_{t-k}(\alpha_t \mathbf{z}_t - \sigma_t \hat{\mathbf{v}}_t^*) + \sigma_{t-k}(\sigma_t \mathbf{z}_t - \alpha_t \hat{\mathbf{v}}_t^*)$
          $\mathbf{z}_{t-2k}^* = \alpha_{t-2k}(\alpha_{t-k} \mathbf{z}_{t-k}^* - \sigma_{t-k} \hat{\mathbf{v}}_{t-k}^*) + \sigma_{t-2k}(\sigma_{t-k} \mathbf{z}_{t-k}^* - \alpha_{t-k} \hat{\mathbf{v}}_{t-k}^*)$
          $\hat{\mathbf{x}}_t^{Target} \equiv \hat{\mathbf{x}}_t = \dfrac{\mathbf{z}_{t-2k}^T - \frac{\sigma_{t-2k}}{\sigma_t}\mathbf{z}_t}{\alpha_{t-2k} - \frac{\sigma_{t-2k}}{\sigma_t}\alpha_t}$               ▷ Target Estimate
          $L_\eta = \max\left(1, \frac{\alpha_t^2}{\sigma_t^2}\right) ||\hat{\mathbf{x}}_t^s(\eta) - \hat{\mathbf{x}}_t^{Target}||_2^2$
          $\eta = \eta - \gamma \nabla_\eta L_\eta$                                     ▷ Optimization
          Update $\mathcal{D}_{DAgger}$ using Algorithm 1
      **end while**
      $\theta \leftarrow \eta$                                           ▷ Update teacher with current student
      $N \leftarrow N/2$                                    ▷ Halve the number of teacher denoising iterations
  **end for**

---

Table 2: Evaluation with different teacher selection rates.

| $p_T$ | FID [↓] | CLIP [↑] |
|---|---|---|
| 0.8 | 23.14 | 0.319 |
| 0.6 | 22.33 | 0.317 |
| 0.4 | 21.95 | 0.313 |
| 0.2 | 21.92 | 0.307 |

Table 3: Switching from $Teacher \rightarrow student$ within single generation.

| $T$ | FID [↓] | CLIP [↑] |
|---|---|---|
| 749 | 23.60 | 0.309 |
| 500 | 23.80 | 0.317 |
| 250 | 24.21 | 0.320 |

Table 4: Switching from $Student \rightarrow teacher$ within single generation.

| $T$ | FID [↓] | CLIP [↑] |
|---|---|---|
| 749 | 21.64 | 0.321 |
| 500 | 21.16 | 0.314 |
| 250 | 22.05 | 0.306 |

to assess if the teacher model can improve student generation from intermediate time steps. We vary the prior probability $(p_T)$ of selecting the teacher model for each (t-250) of transition. Results in Tab. 2 show that decreasing $p_T$ (less frequent teacher usage) leads to a decline in CLIP score, suggesting the teacher model improves student predictions across various time steps. To understand if there are any critical timesteps for teacher intervention, we analyze the impact of switching between teacher and student at specific time steps (Tab. 3 and 4). The results indicate that teacher intervention, either early or late in the diffusion process, can improve generation quality compared to student-only inference. This further supports the presence of covariate shift and its impact on student model.

### E.1 COVARIATE SHIFT VISUALIZATION

## F PROMPT INVERSION

In Table 5, we investigate if the underlying map from 'noise' space to 'data' space is preserved during the distillation phase of diffusion. Updating the map could have an implication on adopting various tools obtained on pre-trained diffusion model with applications in personalization etc. To capture this we consider a setting where we obtain inverted prompts of a reference image using Mahajan et al. (2023) on the COCO dataset and then pass the inverted prompts to distilled models to capture the similarity of the generated image to the reference image and also diversity of generations. As PH2P returns the optimal token for a given image, if a relative change in the map is minimal we expect the generated output to be more aligned with the reference image. If distilled model has good behavior

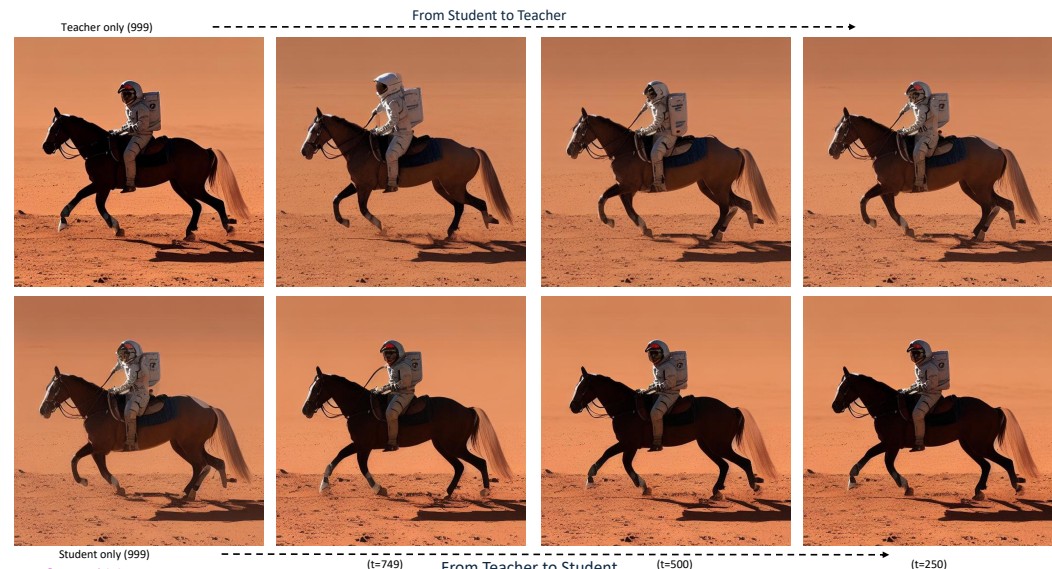

Figure 2: Sensitivity of timestep in reverse process

Table 5: Evaluating baseline based optimized/PH2P prompts on distilled models, showing the effectiveness of map-preserving multi-step distilled methods over other fewer step distillation methods

| Model | Steps | LPIPS_Div |
|---|---|---|
| ADD | 1 | 0.51 |
| LCM | 4 | 0.45 |
| LCM-LoRA | 4 | 0.53 |
| Instaflow(0.9B) | 1 | 0.60 |
| SD | 32 | 0.63 |
| Step Distillation | 8 | 0.61 |
| DDIL | 8 | 0.60 |
| Step Distillation | 4 | 0.60 |
| DDIL | 4 | 0.60 |

we expect distilled models to preserve diversity of generation on invertd prompts too, our overall findings are consistent with text-guided generation for inverted prompts too.

# G    ADDITIONAL QUALITATIVE EXAMPLES

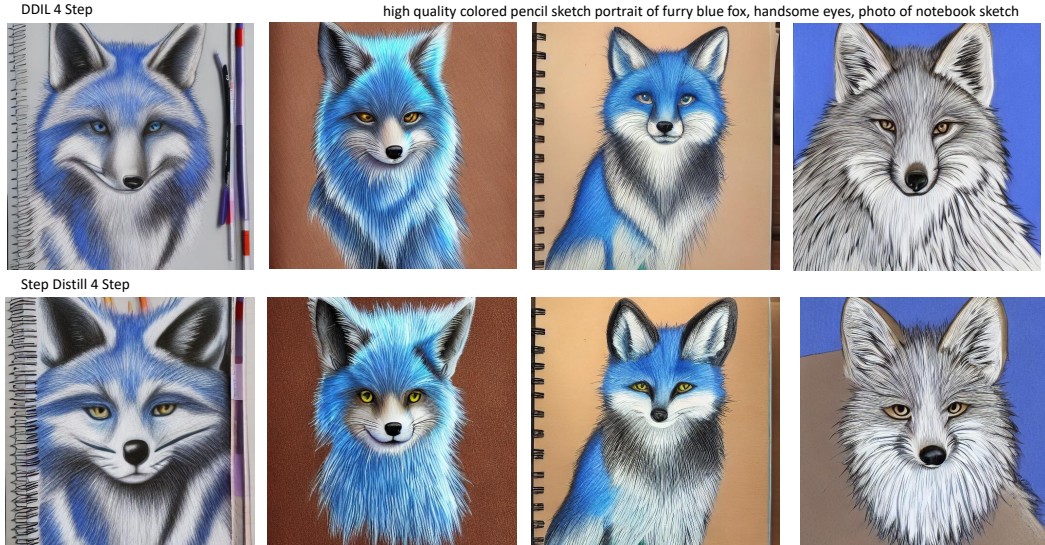

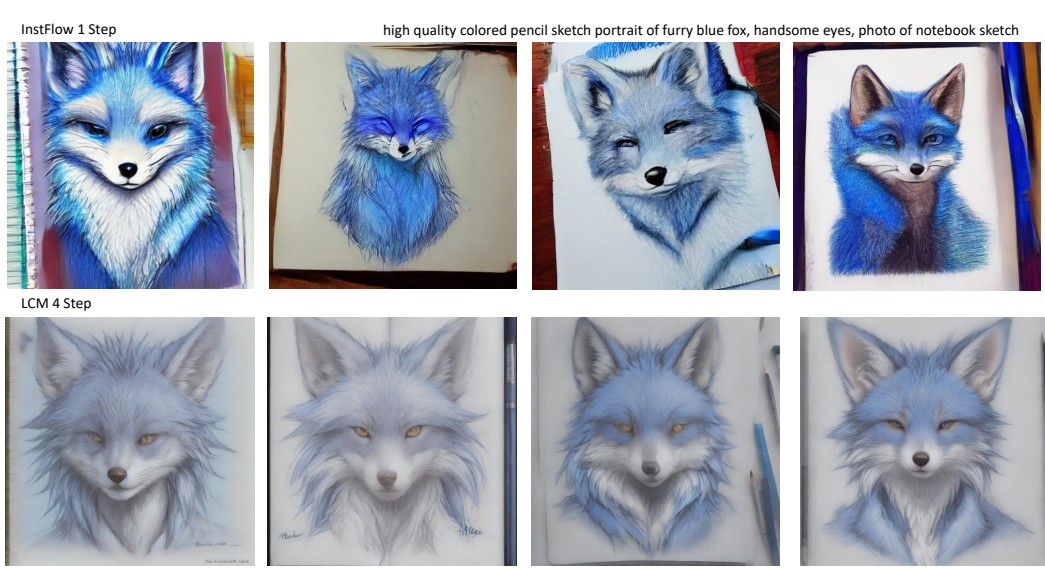