# OpenReview forum: "DDIL: Improved Diffusion Distillation with Imitation Learning"
_ICLR.cc/2025/Conference — ICLR 2025 Conference Withdrawn Submission_

### Official Review · Reviewer_mWY3 · 2024-10-31

**Soundness:** 3
**Presentation:** 4
**Contribution:** 3
**Rating:** 6
**Confidence:** 4

**Summary:**

The paper tackles the challenge of lengthy inference times in diffusion models caused by numerous denoising steps.  While prior works like Progressive Distillation and Latent Consistency Matching address this, they often compromise generation quality. The authors attribute this degradation to covariate shift, where the distribution of noisy input latents encountered by the student model during training deviates from the distribution observed during inference.  To mitigate this, they propose DDIL, a method that enhances the training distribution by incorporating both the data distribution (forward diffusion) and student-induced distributions (backward diffusion), drawing inspiration from the DAgger algorithm in imitation learning. DDIL hypothesizes that training on the data distribution preserves diversity by maintaining the marginal data distribution, while training on the student distribution corrects compounding errors by addressing covariate shift.

I particularly liked the well crafted intuition and parallels to imitation learning, as well as the clear differentiation with prior works in Table 1. Besides, Figure 2 provides a clear and intuitive understanding of the method by demonstrating the teacher and student rollouts and how the losses are computed. The authors tested their method across a couple of prior works in diffusion distillation and have impressive results (improved FID and CLIP score) showing the effectiveness of their approach. While the experiments are only for image generation, the approach is general and could also potentially be applicable to other domains such as audio and video generation thus accelerating research in real-time media generation. The paper could have been improved with some more details and discussion around limitations and future work. Overall, I think the work is novel and well articulated. I would be in favor of accepting the paper with a few asterisks / improvements in the paper that I have listed in the following sections.

**Strengths:**

The paper proposes a novel approach drawing parallels from Imitation learning and specifically Dataset Aggregation (DAgger) inspired training distribution augmentation, which is well tested in literature both theoretically and empirically. The paper conducts thorough experiments on standard benchmarks, comparing DDIL against multiple established distillation techniques. The authors also show that the approach is not just applicable on one method but across a few diverse methods showing it is generally applicable. The results demonstrate consistent improvements in FID and CLIP scores, supporting the effectiveness of the proposed approach. The paper is well-written and organized. The motivation, methodology, and results are presented clearly. The figures are effective to illustrate key concepts. The problem of addressing covariate shift in diffusion model distillation is crucial for improving the efficiency and practicality of these models.

**Weaknesses:**

* **Missing details on Sampling Priors (β):**
   * The algorithm mentions user-defined sampling priors (β_frwd, β_teach_bckwrd, β_student_bckwrd) to control the selection of intermediate latent variables. However, it does not specify how these priors are defined, updated, or scheduled during training. This is a crucial detail for understanding the practical implementation of DDIL and its behavior. The authors should provide more detail / ablations are needed on how these are initialized and potentially adjusted during training (e.g., based on a schedule, the student's performance, etc.) on how changes in these priors impact results.
* **Lack of qualitative discussion:**
    * Figure 1, though included, is not referenced in the paper. The authors should consider elaborating and explaining the qualitative improvements achieved by DDIL across the different scenarios depicted in the figure, focusing on a few specific examples, highlighting how DDIL improves specific aspects of image quality or fidelity compared to the baseline methods. This would strengthen the understanding of its effectiveness and justify the figure's inclusion.
* **Reflected diffusion and Thresholding:**
    * The paper mentions the use of "reflected diffusion" and thresholding for stability. While the concept is explained in the text, the algorithm itself does not show where and how the thresholding is applied (e.g., to the teacher's score estimates, the student's predictions, or both).
        * It would be better if it is also reflected and explained in Algorithm 1 on how it fits into the overall method. While the concept is introduced, the practical details of thresholding are unclear. What specific threshold values are used? How are they chosen or adjusted? The authors should add more details about these questions in the paper and Algorithm 1.
    * While the authors mention improved stability and performance, they lack specific quantitative results isolating the impact of thresholding on covariate shift mitigation. (e.g., measuring gradient variance, loss fluctuations)
* **Limited analysis of diversity preservation:**
    * In Table2, the diversity / LPIPS score for PD does not seem to degrade but that of LCM degrades a bit with DDIL.
    * Is there any insight or empirical results showing if that is always the case for PD?
    * Are these reported results statistically significant?
    * There is a small section in the end talking about diversity vs. quality but I'd encourage the authors to elaborate more on their diversity results including statistical significance tests and how they relate to this tradeoff along with any future considerations, and discussion of why DDIL might affect diversity differently for PD and LCM.



**General suggestions for improvements:**
  * Expanding the discussion on the limitations of DDIL and potential future research directions would improve the completeness of the paper.
  * In Table 2, the authors could break down individual ablations and show delta gains to highlight the effectiveness and changes from their experiments better. Specifically, introducing another small delta column and breaking down the LCM experiments in Table 2 separating them with the PD results below through lines and highlighting the delta with base.
  * The authors should also consider citing: “Photorealistic text-to-image diffusion models with deep language understanding.” during the discussion of reflection / thresholding which was an earlier work to alter the diffusion sampling process through thresholding.



**Things that could be improved but did not impact the score:**


* Typos & Spelling Errors:

    * Line 63: "a white robot, a red robot and a black robot standing together"  ->  "a white robot, a red robot, and a black robot standing together" (missing comma before "and")
    * Line 80:  "motorcyle" -> "motorcycle"
    * Line 110: "on on both" -> "on both" (duplicate word)
    * Line 177: "excarbates" -> "exacerbates"
    * Line 184: "prounced" -> "pronounced"
    * Line 314: "threshold-ed" -> "thresholded"
    * Line 449: "Ss discussed" -> "As discussed"

* Grammar & Clarity Suggestions:

    * Line 42:  "when the number of denoising steps are low" -> "when the number of denoising steps is low"
    * Line 174: "With in iterative" -> "Within iterative"
    * Line 178: "in accumulation of error" -> "in an accumulation of errors"
    * Line 226-233: This paragraph about "Mode Seeking" is somewhat confusingly worded. Rephrasing to focus on the core problem of reduced diversity and its cascading effects would improve clarity.
    * Line 376: "In this work we consider" -> "In this work, we consider"

**Questions:**

* The authors have drawn parallels more towards imitation learning, were any other or more recent SOTA imitation learning methods tried? If not, why? Some considerations are:
    * SafeDAgger,
    * Generative Adversarial Imitation Learning (GAIL): GAIL and its variants (e.g., AIRL (Adversarial Inverse Reinforcement Learning)) that use a discriminator to distinguish between expert and student trajectories. This adversarial training can lead to more robust policies and better generalization. While the paper mentions adversarial losses in the context of distribution matching, applying a GAIL-style approach directly within the imitation learning aspect of DDIL could be interesting.
* Missing details on the sampling priors and how were they chosen?

**Details Of Ethics Concerns:**

Paper is on Arxiv.

---

### Official Review · Reviewer_EBou · 2024-11-01

**Soundness:** 3
**Presentation:** 2
**Contribution:** 2
**Rating:** 3
**Confidence:** 4

**Summary:**

This work addresses the limitations of the score distillation of diffusion models. Specifically, the authors identify the imbalance of mode-covering and mode-seeking behavior due to the current design of the distillation methods, resulting in trade-offs of performance and inference NFEs.

To counter this, they propose Diffusion Distillation in Imitation Learning (DDIL), which trains on both the data reference distribution to preserve diversity and the student-induced reference distribution to address balance of mode-covering and mode-seeking. By combining these distributions, DDIL enhances performance across various distillation methods, including Progressive Distillation (PD), Latent Consistency Models (LCM), and Distribution Matching Distillation (DMD2). The results show that this approach enhances both quality and diversity, providing a more practical solution for efficient diffusion distillation.

**Strengths:**

- The proposed method is overall simple, straighforward but effective. The idea of imitation learning naturally construct the student-induced reference distribution to enhance the mode-seeking behavior in the distillation.

- The paper is overall well-written and easy to read.

- The proposed DDIL method can be naturally introduced in a variety of existing distillation methods.

**Weaknesses:**

- The discussion of co-variate shift in section 3 is fairly weak and the authors lack of corresponding justification of this argument in the following experiments. It is not clear where the co-variate shift exists in each score distillation methods and how does it affect the final performance. Moreover, it is confusing how DAgger approach is related to DDIL and how the use of total variation distance connects the loss function in Eq (2).

- The training on the backward trajectory also requires additional sampling of the trajectory. The authors should discuss the additional cost in this part. Moreover, it is non-trivial to discuss or conduct ablation studies on how the sampled trajectory affects the distillation.

- Considering the default of using only forward or reverse KL divergence in the distillation, there are previous works that inspired by other distribution measure such as combining forward KL with GANs (Kim et al, 2023), Fisher divergence (Zhou et al, 2024). The authors may need to carefully address the differences and connections with these previous methods, and also include them into comparison.

```
Ref:

Kim, Dongjun, et al. "Consistency trajectory models: Learning probability flow ode trajectory of diffusion." arXiv preprint arXiv:2310.02279 (2023).

Zhou, Mingyuan, et al. "Score identity distillation: Exponentially fast distillation of pretrained diffusion models for one-step generation." Forty-first International Conference on Machine Learning. 2024.
```

- The evaluation is only conducted on text-to-image generation, while the performance on unconditional/class-conditional generation remains unknown.


-  The improvement of using DDIL on top of Progressive Distillation and LCM in terms of the generation quality, and the improvement on top of DMD2 in terms of diversity is marginal.

**Questions:**

Please see Weaknesses

---

> ### Author Response · Authors · 2024-11-15
>
> We sincerely thank reviewer **EBou** for the constructive feedback and for acknowledging effectiveness, ease of adoption to different distillation methods.
>
> **Covariate Shift in Score Distillation Methods**
> We agree with reviewer **EBou** that current score distillation methods like DMD2 do not suffer from covariate shift as training is performed on backward trajectories and choice of divergence measures can have different implications in terms of coverage or diversity of student's generations. Our work aims to go beyond the default settings of DMD2 by enriching the training data.
>
>
> **DAgger and DDIL**
> We conceptualize the reverse diffusion process as a Markov Decision Process (MDP) and leverage the distribution-matching perspective of imitation learning to enhance sampling quality. Prior work has established that the Data Aggregation (DAgger) framework minimizes the total variation distance between the state occupancy distributions of the student and teacher. DDIL, by collecting and aggregating a dataset incorporating teacher feedback on both the student's backward trajectory and the data distribution, mirrors the DAgger framework. DDIL objective captures importance of sampling distribution within DAgger framework and demonstrate generality of our approach which is agnostic actual feedback used in distillation method. We operate under the implicit assumption that the DDIL objective also minimizes a total variation bound on the trajectories of intermediate latent variables, further investigation into this relationship warrants future research.
>
>
> **Comparison to existing work, Divergence Objectives:**
> We appreciate your suggestion to explore score identity distillation. While we are unfamiliar with this work, we will certainly investigate it for potential incorporation in future revisions of our manuscript.
>
> You rightly point out that different divergence measures can mitigate mode seeking behavior and enhance the diversity of generated samples. However, as detailed in Section 4.1, our current focus lies on effect of training distribution within the diffusion distillation which is complementary to divergence measure. Combining DDIL with other divergence measures is interesting and will consider in future work. Reviewer o2Tv also suggested an experiment to see if DDIL improves on already distilled checkpoint, to recover lost modes of target distribution we might need to carefully select divergence objective and also training distribution.
>
> While adopting improved divergence measures within DDIL could potentially enhance performance by refining the training distribution, we believe a comprehensive exploration of design choices, including divergence measures, training distributions, noise-level weighting, and feature-space feedback mechanisms, warrants further investigation. These elements are complementary in their effects, and our work highlights the significance of training distribution as a crucial design choice, complementing existing research in this domain.
>
> We acknowledge the potential benefits of incorporating your suggestion and plan to delve deeper into these aspects in future research.
>
>
> **Evaluation**: Regarding your suggestion for class-conditional generation evaluation, we are open to exploring this avenue. To better understand your perspective and potentially refine our future research directions, we would appreciate it if you could elaborate on the specific novel insights you anticipate from a class-conditional evaluation.

---

### Official Review · Reviewer_jXnp · 2024-11-03

**Soundness:** 2
**Presentation:** 3
**Contribution:** 2
**Rating:** 3
**Confidence:** 4

**Summary:**

This paper tries to solve the diffusion models’ limitation of multiple denoising network passes and identify the co-variate shift as one of the main reasons. To address co-variate shifts and achieve a better performance of diffusion models, the authors proposed Diffusion Distillation with Imitation Learning (DDIL). DDIL draws inspiration from the Dataset Aggregation (DAgger) method and alleviates the co-variate shift by incorporating the diffusion distillation process into an imitation learning framework. It trains the student model on the data distribution (forward diffusion) and the distribution generated by the student model (backward diffusion). Besides, the authors introduce the reflected diffusion formulation and apply thresholding to the score estimates of both teacher and student models.

**Strengths:**

The research direction is relatively interesting, and the review of related work is thorough.

**Weaknesses:**

1. In Line 38, the authors claim that 'Multi-step student models offer a promising approach in balancing quality and computational efficiency. However, they often face a critical challenge: covariate shift.' However, there is no theoretical or experimental evidence provided to substantiate the covariate shift issue. Please validate the 'covariate shift issue', such as visualizing the potential distribution of different time steps or measuring the distribution offset between training and inference, etc. Besides, the experimental results only show improvements in quantitative metrics for generated images with the proposed method, without directly addressing or resolving the covariate shift problem.

2. Commonly used metrics, such as the Inception Score, and additional datasets beyond COCO, such as CIFAR100 and ImageNet, etc., should be employed to validate the experimental conclusions drawn in this paper.

3. I can not conduct a comprehensive review of the technological accuracy of this paper, as it presents an empirical rather than theoretical approach, and the implementation code is not provided (particularly the implementation code of Algorithm 1).

**Questions:**

Please refer to the Weaknesses and Questions above.

---

> ### Author Response · Authors · 2024-11-15
> **Rebuttal by Authors**
>
> We thank reviewer **jXnp** for feedback.
> We hope our global response addresses reviewers concerns on method complexity, empirical results and covariate shift justification.
>
> As discussed in our background section, we see reverse diffusion process as an MDP and adopt distribution matching view of imitation learning to improve sampling distribution in our work. As we are not proposing novel algorithm but rather showing equivalence in problem settings and assumptions, we did not see need for more extensive review of background theoretically as DDIL is more of a framework building up on previous work which we discussed, we will work on improving presentation and formulation in next version.
>
> **Evaluation**
> We appreciate the reviewer's suggestion to evaluate our method on smaller datasets like CIFAR100. However, we believe our focus on large-scale models like SDv1.5 and SSD1B/SDXL is justified for several reasons.
>
> Firstly, our method is designed to address the challenges specific to training and fine-tuning massive language models. These challenges, such as computational cost and data efficiency, are significantly more pronounced in large-scale settings. Evaluating on smaller datasets might not accurately reflect the performance and benefits of our method in real-world applications.
>
> Secondly, our evaluation on two distinct models and three distillation methods already provides a comprehensive assessment of our method's generalization and robustness. Expanding to smaller datasets would add complexity without necessarily yielding novel insights directly relevant to our target domain.
>
> We are open to exploring the potential benefits of evaluating on smaller datasets in future work. However, for this particular study, we believe our focus on large-scale models is appropriate and provides valuable contributions to the community.
>
> We encourage the reviewer to elaborate on the specific novel insights they anticipate from a CIFAR100 evaluation, as this would help us better understand their perspective and potentially refine our future research directions.

---

### Official Review · Reviewer_o2Tv · 2024-11-03

**Soundness:** 3
**Presentation:** 2
**Contribution:** 2
**Rating:** 5
**Confidence:** 4

**Summary:**

This paper identifies covariate shift along the sampling trajectory as a significant challenge in diffusion model distillation. To address this issue, the authors propose DDIL, a method that enhances the training distribution of noisy latents through three distinct sources: (1) forward diffusion of real images, (2) denoised latents produced by the teacher model, and (3) latents generated by the student model. By incorporating teacher feedback into the student trajectories, DDIL effectively mitigates covariate shift that may result from errors or suboptimal score estimates of the student model, particularly during the initial steps.

**Strengths:**

- Novelty: The paper highlights that errors accumulated during the early stages of sampling can exacerbate covariate shift and introduce potential biases. Section 3.2 and Figure 2 effectively illustrate this issue and demonstrate a solution through querying the teacher model during the reverse sampling process. The construction of a teacher-student symbiotic training dataset is a novel approach that could generate beneficial synergies.

- Quantitative Results: The effectiveness of DDIL is validated using three state-of-the-art distillation models—LCM, Progressive Distillation, and DMD2—that represent commonly used groups in distillation modeling, whether directly predicting the ODE endpoint or the endpoint of sub-intervals.

**Weaknesses:**

- Lack of Empirical Analysis & Ablation Study: While the proposed “Teacher Backward” mechanism represents a novel component in Algorithm 1, the use of (2) real images and (3) student backward methods are less innovative, with student backward resembling a consistency constraint. Thus, an ablation study isolating the impact of Teacher Backward is essential to validate the framework. For instance, what results emerge when comparing the use of (2) real images + (3) student backward against the combined approach of (1) Teacher Backward + (2) + (3) under a fixed computational budget? Although there is some analysis related to covariate shift in Tables 2, 3, and 4 in the appendix, it primarily addresses stochastic mixing of teacher and student outputs during inference. Additionally, these tables are difficult to interpret due to conflicting trends between FID and CLIP scores; the authors prioritize CLIP scores, basing claims on their trends. A more thorough examination of the Teacher Backward component in DDIL is warranted.

- Metrics & Experimental Settings: Given that traditional metrics like FID and CLIP scores may not fully capture human preference or image quality, recent distillation studies often incorporate alternative metrics for assessing high-frequency details or human preference (e.g., Image reward [1], Patch FID [2], etc.). Such metrics are required here, as the DDIL improvements is apparently marginal in the CLIP and FID trends. Furthermore, the qualitative results (e.g., Figure 1 and related figures in the appendix) do not convincingly demonstrate improvement. Additional comparisons and a large-scale user study with uncurated samples would further support the findings.

- Writing Quality: The overall writing quality is subpar, with numerous typos, incomplete discussions, and key quantitative results relegated to the appendix despite available space in the main text. For instance, the discussion of the diversity-quality tradeoff in Line 447 is vague—does unrolling with the teacher improve or impede diversity? It appears that DDIL reduces diversity based on $LPIPS_{Diversity}$, despite using a teacher network. Additionally, baseline (student-only) results are missing in Tables 3 and 4 in the appendix, yet the authors assert that teacher intervention improves quality. A precise mathematical definition of covariate shift would help reader comprehension, as interpretations may vary from dataset-related shifts to time-dependent sampling path discrepancies.

---
**References**

[1] Xu, Jiazheng, et al. "Imagereward: Learning and evaluating human preferences for text-to-image generation." Advances in Neural Information Processing Systems 36 (2024).
[2] Lin, Shanchuan, Anran Wang, and Xiao Yang. "Sdxl-lightning: Progressive adversarial diffusion distillation." arXiv preprint arXiv:2402.13929 (2024).

**Questions:**

Overall, the paper presents a novel concept with promising quantitative results; however, the quality and structure could be refined. Below are some example questions highlighting areas of weakness and corresponding suggestions (please refer to weakness part for full details):

- Ablation Study: How significantly does the Teacher Backward mechanism enhance distillation performance? What is the associated increase in computational cost, and does this approach reduce the diversity of outcomes?

- Metric Trends: Have you considered evaluating performance using additional metrics, such as ImageReward or Patch FID? Can you explain why CLIP and FID trends are contradictory in some tables?

- Alternative Approaches: The authors utilize teacher intervention to improve the training distribution. Have you explored the possibility of employing pre-trained distillation models, such as DMD2 or SDXL-lightning, in place of the teacher for unrolling? Since student models often approximate the integral over sub-intervals, they could also potentially rectify sampling trajectories and mitigate covariate shift but at a lower computational cost.

---

> ### Author Response · Authors · 2024-11-15
> **Rebuttal by Authors**
>
> We sincerely thank reviewer **o2TV** for the constructive feedback and acknowledging novelty of co-variate shift in diffusion distillation, and  construction of a teacher-student symbiotic training dataset within our DDIL framework and resultant performance improvement across methods!
>
> **Ablation with teacher only backward**
> Thank you for interesting perspective and thoughtful question. Motivation of our
> approach is to obtain teacher's **corrective feedback** on student's mistakes and reduce covariate shift across steps as illustrated in Fig 2(b) . So if we assume pretrained diffusion model is a reasonable proxy to real data distribution, training only on teacher would be equivalent to training on data distribution only and hence we did not consider it in current experiments.
>
> Though it is worthwhile to note that any discrepancy in data distribution and aggregate posterior predictive distribution of pre-trained diffusion models can also be minimized during diffusion distillation with right sampling distribution and divergence measure to recover lost modes.
>
> **Evaluation**
> Thank you for suggesting PatchFID will include results in next revision of manuscript. Though we did not consider ImageReward, we report HPSv2, which is trained on much richer training set comparisons in terms of prompts, generative models considered etc. We addressed figure and qualitative comments in our response to reviewer 'jluo'
>
>
> Thank you for suggesting on lack of clarity w.r.t covariate shift within time-dependent sampling path. Will clarify that and improve manuscript in next revision in terms of discussion, and reorganization as suggested.
>
>
> **Why are FID and CLIP contradictory?**  We observe that FID has its deficiencies and sometimes has negative correlation to generation quality and human preference, same observation is made in Pick-Score[1]. Overall we observe CLIP score to be more informative and aligned with human preference especially when FID is comparable in case of SDv1.5.
>
>
> **Fine-tune existing distilled checkpoint**
> It is an interesting suggestion to consider pre-trained checkpoints and see if we can obtain corrective feedback by rectifying sampling trajectories. Though we did not report, we conducted such experiment as initial proof of concept with a pre-trained progressive distillation checkpoint and observe performance boost as expected.
>
>
> [1] Kirstain et al. Pick-a-Pic: An Open Dataset of User Preferences for Text-to-Image Generation

---

### Official Review · Reviewer_jLUo · 2024-11-04

**Soundness:** 2
**Presentation:** 2
**Contribution:** 1
**Rating:** 3
**Confidence:** 4

**Summary:**

This paper aims to address the covariate shift issue in diffusion distillation by employing data augmentation techniques. The authors propose utilizing samples from the data distribution, as well as from the teacher and student model distributions, during the distillation process. Additionally, a thresholding technique from prior work is incorporated to enhance training stability. While the proposed approach is straightforward, the empirical results do not demonstrate significant improvements.

**Strengths:**

1. The proposed method is conceptually simple and easy to understand.

**Weaknesses:**

1. **Figure 1 Analysis**: The visualization in Figure 1 does not convincingly show the superiority of the proposed method. The color scheme and styling are too similar, making it difficult to distinguish improvements. Additionally, the minor differences in output images could potentially be explained by different random seeds.
2. **Covariate Shift Justification**: The explanation regarding covariate shift due to the accumulation error in the backward diffusion process is unconvincing. The paper would benefit from citing relevant literature that supports this claim.
3. **Method Complexity**: The proposed method, which introduces data augmentation into the diffusion distillation process, lacks sufficient novelty for an ICLR submission. The primary contribution—utilizing samples from the data, teacher, and student distributions—resembles more of a practical trick than an innovative approach.
4. **Empirical Results**: The empirical performance is underwhelming, with improvements that are often marginal. Although the paper claims enhanced generation diversity, the results frequently indicate a decline in diversity.
5. **Writing Quality**: The manuscript contains numerous typos and grammatical errors, indicating a lack of careful proofreading.

typo:
1. Abstract: "We show that DDIL **consistency** improves on baseline"
2. Line 108: We propose a novel DDIL framework which enhances training distribution of the diffusion distillation within the dataset aggregation ‘DAgger’ framework by performing distillation **on on** both the data distribution (forward) ...
3. Line 176: which is classic feedback loop **[1]?** in imitation learning
4. ...

**Questions:**

See Weaknesses.

---

> ### Author Response · Authors · 2024-11-15
> **Rebuttal by Authors**
>
> We thank reviewer **jLUo** for feedback.
> We hope our global response addresses reviewers concerns on method complexity and empirical results and covariate shift justification.We will work on manuscript to improve writing for next revision.
>
>
> **Figure 1 analysis**
> We generate figure 1 on same machine with same seed, prompt across different settings of DMD2 starting with pretrained checkpoint as our goal is to qualitatively see effect of distillation method, when controlled for prompt, seed, and training dataset. To perform apples to apples comparison we show equivalent images w.r.t baseline and our contributions, we observe more variance in image layout with change in seeds.Though we agree there is no reason for generation to aligned when distilled within DMD2 objective we observe similar generations for same seed across different methods.

---

### Author Response · Authors · 2024-11-15
**Global Rebuttal by Authors**

We thank the reviewers for their thoughtful and constructive feedback. We are glad reviewers recognize the positive impact of our proposed DDIL framework to mitigate the challenges of covariate shift in diffusion distillation. and construction of a teacher-student symbiotic dataset within our framework!

**Novelty/Innovation**
Goal of our work is beyond simply enhancing the performance of distilled diffusion models. We identify covariate shift in the sequential setting as a key contributor to the performance limitations of multi-step distilled models. This aligns with the observations made in [2] regarding the training-inference mismatch in diffusion distillation. To address this challenge, we leverage the DAgger framework, effectively transforming the sequential learning task into an i.i.d. setting within an online or interactive learning framework which aligns with simulation-free training setup of diffusion models. DAgger's proven sample efficiency, particularly in handling significant distribution shifts, makes it a powerful framework to be considered with diffusion models.

Furthermore, we show connection between diffusion models, sequential decision-making and interactive learning. This novel perspective opens up exciting avenues for future research. We envision exploring data efficiency, continual learning, preference learning, and offline reinforcement learning in the context of diffusion models and distillation
and also potential applications in decision-making especially when diffusion policies are employed. We believe this work represents a valuable scientific contribution and anticipate further research.

**Performance Improvement**
While performance improvement is not our sole objective, our method consistently surpasses state-of-the-art distillation techniques with 12% improvement in FID and improvement on CLIP score, HPSv2 without loss of diversity in case of DMD2. This is particularly noteworthy given the already high performance of distilled models, which often rival their teacher models demonstrating limited headroom for improvement and our approach demonstrates generalization across various methods.

We acknowledge that there is inherent trade-off between quality and diversity in distilling generative models. However, our experimental results demonstrate that DDIL consistently improves performance without sacrificing diversity. Notably, DDIL achieves better diversity than the SSD1B teacher model, even with a significantly smaller batch size (7 compared to 128 in the original DMD2 training setup). This highlights ability of DDIL to retain the teacher's diversity while achieving superior performance with **minimal computational overhead**. We achieve this by collecting sampled intermediate time-step latents into a dataset and updating this dataset at a significantly lower frequency compared to gradient feedback.

**Covariate shift Justification**
We thank reviewers feedback and will try to further clarify covariate shift in diffusion models in section 3.2.

In Table 2 in appendix, we stochastically choose between teacher's higher number of iterations or single iteration of student model for generation within mixed-rollout setting with state-less DDIM Scheduler. Table 2 reveals that decreasing teacher intervention leads to a decline in CLIP score, suggesting the teacher corrects accumulating errors in the student's predictions. In a scenario without covariate shift, performance should be invariant to rate of teacher intervention. However, we observe a consistent improvement in both CLIP score and FID when increasing probability of choosing teacher for transition. This result strongly supports the presence of covariate shift in distilled diffusion model's trajectory, and highlighting the teacher's corrective role.

If a student has a discrepancy compared to teacher for one-transition that discrepancy grows in next transition, i.e., error between teacher and student's latents compounds and grow exponentially. Student's current predictions determines what the student (learner) sees in next step within sequential setting, which is classic feedback loop [1] in imitation learning. This results in covariate shift between training time (forward diffusion/teacher's backward trajectory) latent input  and latents student model encounters when it is unrolled in iterative fashion at inference time. This change in distribution of 'input latents' the student model sees at inference time compared to training time within step distillation or consistency distillation or other methods within multi-step distilled student model is what we refer to as 'covariate shift' that effects generation quality.

References:

[1] Spencer et al. Feedback in Imitation Learning: The Three Regimes of Covariate Shift.

[2] Kohler et al. Imagine Flash: Accelerating Emu Diffusion Models with Backward Distillation

---

### Note · Authors · 2024-11-15

**Comment:**

We thank the reviewers for their time and valuable feedback. We appreciate the constructive suggestions and will incorporate them into future work.

However, due to concerns outlined in our general response, which include disagreements with certain aspects of the reviews, we have decided to withdraw this submission after careful consideration.

**Withdrawal Confirmation:**

I have read and agree with the venue's withdrawal policy on behalf of myself and my co-authors.